# Autonomous Operation Control of IoT Blockchain Networks

**Jae-Hoon Kim [1,2], Seungchul Lee [1,2] and Sengphil Hong [3,*]**

1. Department of Industrial Engineering, Ajou University, Suwon 16499, Korea; jayhoon@ajou.ac.kr (J.-H.K.); ooo890@ajou.ac.kr (S.L.)
2. Department of AI Convergence Network, Ajou University, Suwon 16499, Korea
3. Hancom WITH, Sungnam 13493, Korea
*. Correspondence: sengphil@hancomwith.com; Tel.: +82-10-5348-7772

**Abstract:** Internet of Things (IoT) networks are typically composed of many sensors and actuators. The operation controls for robots in smart factories or drones produce a massive volume of data that requires high reliability. A blockchain architecture can be used to build highly reliable IoT networks. The shared ledger and open data validation among users guarantee extremely high data security. However, current blockchain technology has limitations for its overall application across IoT networks. Because general permission-less blockchain networks typically target high-performance network nodes with sufficient computing power, a blockchain node with low computing power and memory, such as an IoT sensor/actuator, cannot operate in a blockchain as a fully functional node. A lightweight blockchain provides practical blockchain availability over IoT networks. We propose essential operational advances to develop a lightweight blockchain over IoT networks. A dynamic network configuration enforced by deep clustering provides ad-hoc flexibility for IoT network environments. The proposed graph neural network technique enhances the efficiency of dApp (distributed application) spreading across IoT networks. In addition, the proposed blockchain technology is highly implementable in software because it adopts the Hyperledger development environment. Directly embedding the proposed blockchain middleware platform in small computing devices proves the practicability of the proposed methods.

**Keywords:** internet of things; lightweight blockchain; deep clustering; GNN

## 1. Introduction

The enormous volume of data that is generated, collected, and utilized has changed the modern industry. Data now facilitate digital transformations and act as essential factors that promote the convergence of virtual spaces into physical spaces. The blockchain is a key enabler for the implementation of transparent and reliable data transactions. Blockchain technology provides a decentralized system that guarantees data reliability across industrial domains. Unfortunately, common blockchain networks, such as Ethereum or EOS, have an inherent problem in targeting network nodes with considerable computing power. A full node (a device that validates transactions) usually owns a copy of the entire blockchain, which also contains user accounts and balances. However, the IoT network has significant limitations in adopting common blockchains; the requisite amount of computing power is not available in IoT nodes. Energy consumption is another challenge in IoT devices powered with batteries. For IoT blockchain networks, typical cloud-centered IoT architectures have inherent vulnerabilities [1], having the cloud as a point of failure. The fog or edge computing architecture offloads processing from the cloud to the edge of the network. This type of architecture allows for multi-layered networking of IoT devices. In our previous work [2], we reorganized the blockchain software structure and developed new software modules, such as the consensus engine, validator, database, and data-serialization functions, for small-scale IoT development devices. The developed blockchain software module embedded in IoT sensor devices guarantees secure data transactions for distributed

applications. The IoT device builds blocks and performs a consensus as a blockchain validator. We proved that the proposed lightweight blockchain software module can enable blockchain solutions for IoT applications.

However, these architecture designs, software modules, and efficient communication methods are not sufficient for the practical operation of IoT blockchain networks. The most important characteristic of IoT networks is the behavior of dynamic network nodes over time. IoT nodes continuously change their positions and generate transactions with variable rates. We suggest two essential operation strategies to cover dynamic node behaviors in IoT blockchain networks: (1) dynamic IoT blockchain network configuration through deep clustering and (2) reduced dApp (distributed application) spreading through graph neural network (GNN) node classification. Recent machine learning frameworks, such as deep clustering [3] and GNN [4], provide remarkable methodologies for IoT blockchain networks. The most important advantage of neural network is its usability under insufficient information. Some IoT networks have operation instability, then the IoT node behavior data can be hard to be harvested. The neural networks embedded in the proposed deep clustering can provide extracted features even with insufficient behavioral data of IoT nodes. The test simulation environment shows the practical applicability of the proposed operation control methods. We built a cloud-based test environment using our custom-developed blockchain software modules. The multiple virtual machines in the cloud employ the blockchain software module using the Docker container. The superior performance of our approach is effectively illustrated using the developed test environment.

## 2. Related Works

The use of decentralized systems has been suggested to create peer-to-peer IoT applications [5–7]. Blockchain has been effectively applied to IoT applications [8], such as monitoring [9,10], data storage [11,12], identity handling [13], timestamping [14], living services [15], transportation [16], wearable devices [17], supply chains [18], crowd sensing [19], law [20], and security in mission-critical environments [21]. Blockchain technologies can also be used in IoT agricultural applications. Tian [22] presented a traceable application for the supply tracking of agricultural products. The application uses radio frequency identification (RFID) chips and a blockchain to enhance food safety and quality while reducing losses in logistics. Other researchers have provided an IoT device management solution using a blockchain [23]. Researchers have also proposed a system for remote control of IoT devices. The system stores public keys in a public blockchain, such as Ethereum, while saving private keys to each IoT device.

A multi-layer IoT architecture that deploys blockchain technology is described in [24]. The proposed architecture reduces the complexity of deployment by constructing a multi-level IoT configuration and applying the blockchain to each level. A slightly different approach is presented in [25]. This work evaluated the use of a cloud and fog computing architecture to provide blockchain applications. The researchers evaluated the empirical performance of the architecture by using IoT nodes based on Intel Edison boards and IBM's Bluemix as blockchain technology. Further, they noted that it is difficult to apply a regular blockchain such as Ethereum or EOS on traditional resource-constrained IoT nodes. Software defined networking (SDN) has also been suggested for deploying blockchain to IoT architectures. One novel blockchain-based architecture used SDN to control the fog nodes of an IoT network [26]. Another lightweight architecture compressed the communication overhead introduced by the use of a blockchain [27,28]. This work employed an overlay network and a private ledger to replace regular blockchain technology.

Optimized network configuration plays a significant role in lightweight blockchain operation by increasing the efficiency of transaction exchange within the blockchain network. To increase transaction exchange efficiency, the blockchain network node compresses transaction data between nodes (e.g., Compact Block [29], Xthin Block [30], Graphene [31]) or uses a relay facility dedicated to data processing within the network (e.g., FIBRE [32], FALCON [33]). As the above method was intended for a large-scale public blockchain

network on a global scale, it is not an effective method in the context of a private blockchain network in an IoT environment. IBM's ADEPT [34] model presents a decentralized network structure in the IoT environment; however, it predefines the role of nodes, which restricts the flexibility of network configuration and operation in an IoT network where the nodes are frequently moved and relocated. Ardor [35] represents the regional proliferation and activation of dApps to prevent performance degradation to a part of the network while ensuring transaction data integrity.

Ahmed et al. [36] and Khan et al. [37] illustrate the reliability and security concerns in the current IoT environment. Ahmed et al. [36] suggests a comprehensive and systematic mapping study to IoT structure. It categorizes the research evidence for IoT quality assurance appeared. Khan et al. [37] outlines the security requirements for IoT and the state-of-the-art of security solutions. They discuss how blockchain can be a key enabler to solving many IoT security problems.

## 3. Dynamic IoT Blockchain Network Configuration

Dynamic network clustering is an essential technique for flexible blockchain network configuration in an IoT environment. Clusters of nearby nodes are formed from the position information of the network nodes, and the cluster is managed by assigning a chain ID (e.g., chain #:validator/client/center location/pubsub period) (see Figure 1a). The publish/subscribe (pub/sub) transaction exchange is the basic method of communication in the IoT blockchain cluster. Transactions generated by network nodes are propagated using the pub/sub method (e.g., MQTT [38], Kafka [39]). Figure 1b shows the pub/sub transaction exchange method between blockchain network nodes in an IoT environment. When the client node N5 creates a transaction and publishes it to the connected pub/sub broker (N1), the validators (N1, N2, N3, and N4) forming the chain receive the transaction through a subscription to N1. Then, the validators record the hash value of the exchanged transactions to the distributed ledger.

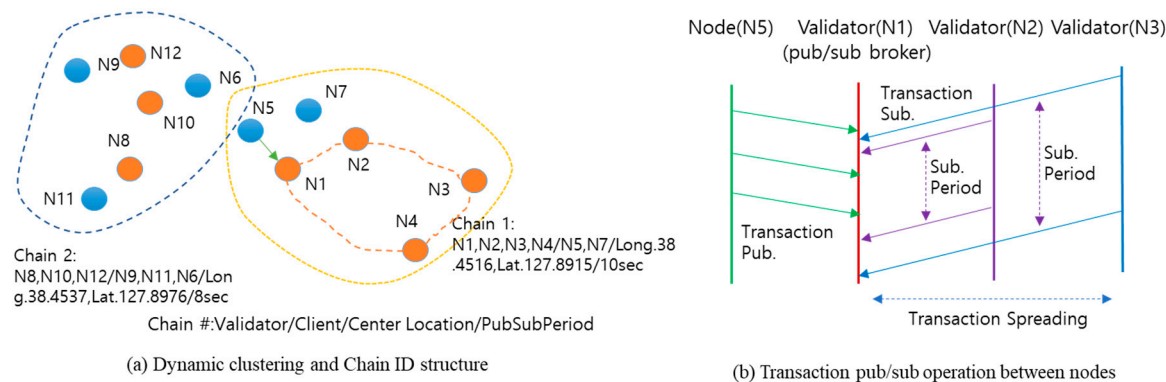

(a) Dynamic clustering and Chain ID structure

(b) Transaction pub/sub operation between nodes

**Figure 1.** Dynamic blockchain network clustering: (**a**) dynamic clustering and Chain ID structure; (**b**) transaction pub/sub operation between nodes.

Deep clustering [3] is a clustering method that jointly trains the parameters of a deep neural network and generates clusters of objects. Combined with a standard clustering algorithm, such as k-means clustering, it iteratively groups the objects (more precisely, it groups the coded features of objects) and uses the subsequent clustering as supervision to update the weights of the feature extractor. Originally, deep clustering was applied to image grouping (i.e., images are handled as objects). When there are no preliminary group labels assigned to images, deep clustering is useful for assigning appropriate labels to images. A feature extractor generates the coded features of images, and standard clustering creates image groups using the features of images. The pseudo labels are obtained from image groups (i.e., the pseudo label $y_{ij} = 1$ when the image $i$ is assigned to group $j$). For each iteration, the pseudo label is used to evaluate the correctness of the feature

extractor (usually with a convolutional neural network). The parameters in the feature extractor are updated to generate better features for images.

Deep clustering has greater flexibility to represent network node behaviors. The node behavior (e.g., positions, generated transactions, the destination of transactions) can be recoded to time series data. A two-dimensional tensor is suitable to record the time-varying behavior of each network node. Then, a set of 2D tensors is used to fully represent node behaviors in a network (see Figure 2). Node behaviors are recoded to the expandable 2D tensor, and recent node behaviors within a time window are collected to build a network behavior snapshot.

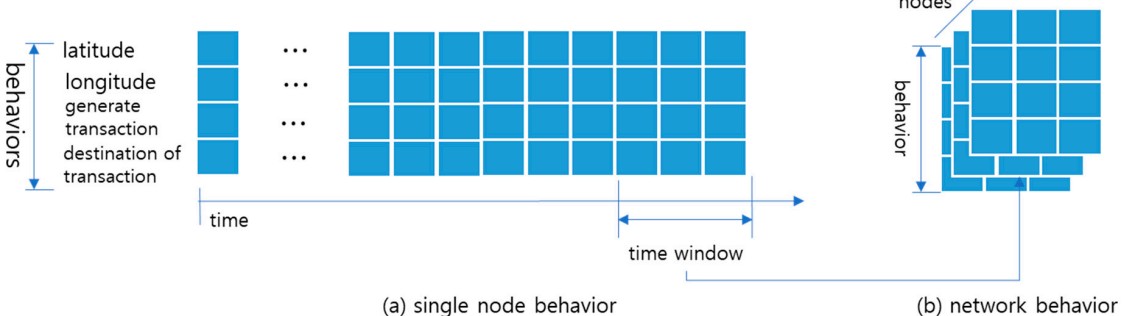

**Figure 2.** A 2D tensor form presenting node behavior: (**a**) single node behavior; (**b**) network behavior.

The feature extractor uses a network behavior snapshot as input data, and a feature vector obtained from the extractor is used to build network node clusters. A standard clustering method, such as k-means, generates clusters. Network clusters provide pseudo labels to their member nodes. The pseudo label is given as a 1D tensor, and the labels of all network nodes are used to evaluate the quality of feature extraction (see Figure 3). The generated features are continuously enhanced by the iterative operation of feature extractor and standard clustering method. The standard clustering method uses the current features and provides pseudo labels for IoT node clustering. The result of node clustering is used to train the feature extractor.

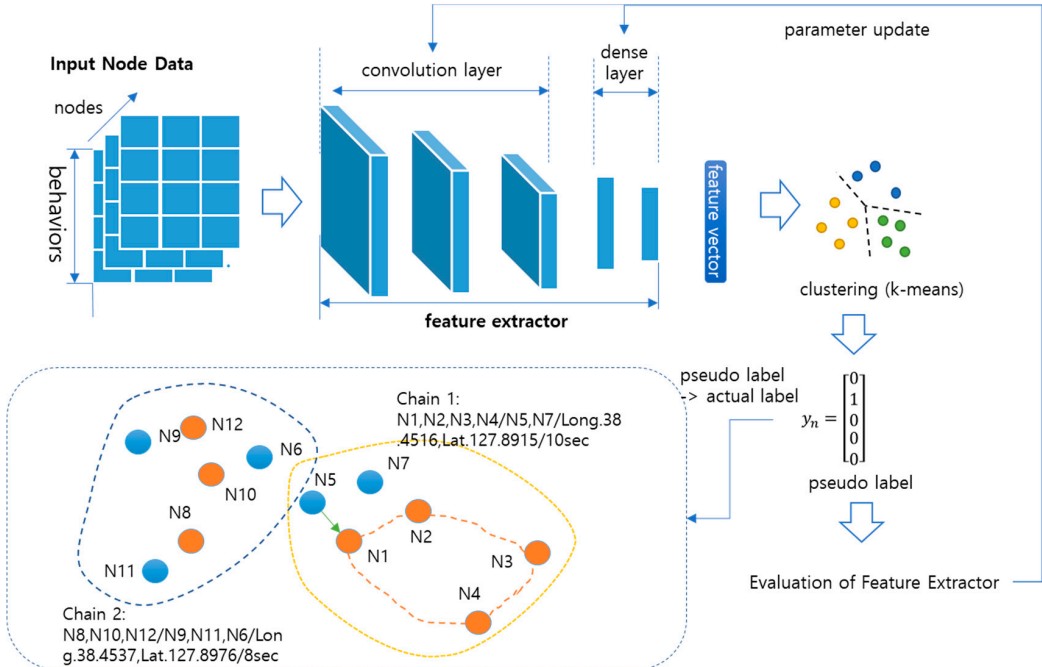

**Figure 3.** Deep clustering for dynamic IoT blockchain network.

Equation (1) evaluates the quality of feature extraction. $f_\theta$ denotes the feature mapping of node behavior $x_n$. We want to find a parameter $\theta^*$ such that the mapping $f_{\theta^*}$ produces good general-purpose features. A parameterized classifier $g_W$ predicts the correct labels on top of the features $x_n$ (note that a *softmax* classifier is typically used). The parameter $W$ of the classifier and parameter $\theta$ of the mapping are then jointly learned by minimizing Equation (1). $l$ evaluates the difference between the prediction ($g_W$) and pseudo label ($y_n$). $L$ denotes the summation of $l$ for all network nodes.

$$L = \sum_n l(g_W(f_\theta(x_n)), y_n) \tag{1}$$

The proposed deep clustering adopts the Convolutional Neural Network (CNN) to extract the behavioral features of IoT nodes. CNN learns the filters automatically without mentioning them explicitly. These filters help in extracting the spatial and sequential features from the input data. The gathered data from the IoT node have spatial cues and sequential information. The position (i.e., longitude and latitude) data have a time-consecutive feature. The destination of transactions provides the spatial information of the network structure. Moreover, the advantage of deep clustering is from the benefit of the CNN structure itself. The findings of Caron et al. [3] illustrate the structural advantage of CNN in deep clustering. Even the untrained CNN adopted in deep clustering provides generally acceptable performance of object clustering.

## 4. Reduced dApp Spreading Using GNN Node Classification

Blockchain networks spread distributed applications across the network, execute them, and verify the results. The holistic spread of distributed applications across the network guarantees transaction data integrity but is also a major cause of performance degradation of the blockchain. To implement a lightweight blockchain, an efficient method of dApp spreading is required. Optimized dApp spreading can significantly improve processing speed during the transaction verification process (for example, peer review or result validation), which is mainly performed by the validators. Regional proliferation and activation of dApps is one way to limit performance degradation to a part of the network while ensuring transaction data integrity. Ardor [36] represents one such approach. However, Ardor currently remains limited to the idea of restricting dApps in large and static networks. We propose a dApp spreading method that can be linked with dynamic clustering in an IoT environment. In the network cluster, a software agent powered by artificial intelligence assigns tags (spread/skip/activation) to nodes according to their states and computing loads (see Figure 4). The total amount of computing and communication loads in the cluster determines the spreading speed of dApps.

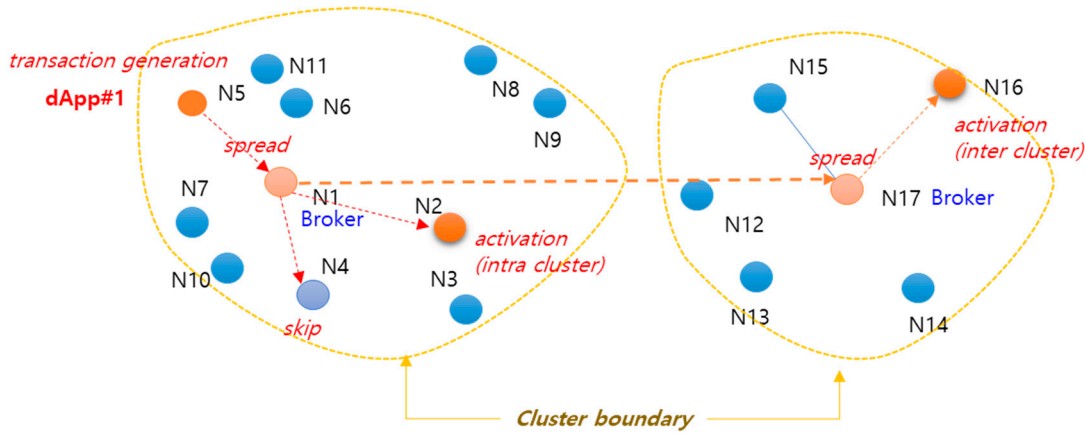

**Figure 4.** Regional dApp spreading and execution.

GNNs are a type of artificial neural network used in graph structures [4]. Commonly known artificial neural networks include convolutional neural networks (CNNs) or recurrent neural networks (RNNs). These artificial neural networks typically use vector or matrix forms as the input data, whereas in the case of a GNN, the input is a graph structure. GNNs can be used for node classification problems in a graph. Each node in a blockchain cluster has a tight relationship with its adjacent nodes; this mutual influence between nodes is important for the regional spread and activation of dApps. Learning in the blockchain network clustering shares many ideas with the node classification by GNNs. For example, we can use cross entropy loss as the loss function (*l*) for the proposed GNN. In addition, we can apply batch training for all nodes in one cluster.

First, we define the behavior embedding structure of nodes. The embedding tensor described in Section 2 has a relatively simple form to present node behavior (note that we focus on the time-varying behavior of nodes in Section 2). We expand the behavior structure to address node capability and current status, such as computing power, communication load, memory, ledger size, and the number of stored dApps. In addition, the feature is specified for each time epoch. We track the time-specific behaviors of nodes using the time index. The GNN has aggregation (AGGREGATE) and concatenation (CONCAT) functions to extract the feature vectors of nodes. The input behavior tensor of a node is concatenated with its adjacent nodes' aggregated behaviors (see Figure 5). The iterative application of AGGREGATE and CONCAT functions to each node in the cluster can determine the features of all nodes in the cluster (see Box 1). Then, nodes are classified based on the features obtained from AGGREGATE and CONCAT. The following pseudocode describes feature extraction using the AGGREGATE and CONCAT functions:

**Box 1.** Aggregation and Concatenation.

```
for v ∈ V:        #V is node set in cluster
    x_v = behavior tensor of node v
    f_v^0 = x_v initial feature of node v
for k = 1 to K:
    for v ∈ V:
        a = AGGREGATE(f_u^{k-1} | (u,v) ∈ E) #E is edge set in cluster
        f_v^i = CONCAT(f_v^{k-1}, a)
```

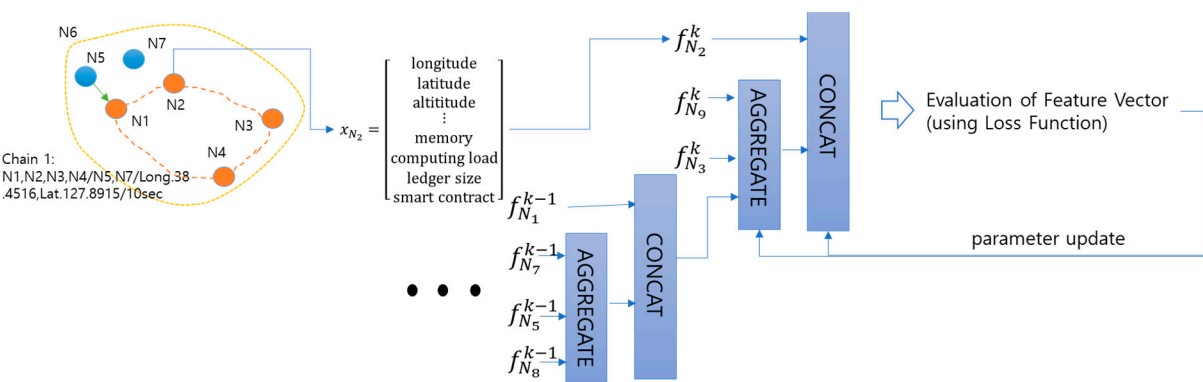

**Figure 5.** Example of GNN training.

$f_v^k$ denotes the extracted feature of node *v* at iteration *k*. The AGGREGATE function simply averages the adjacent nodes' features ($f_u^{k-1}$, $(u,v) \in E$) at the previous iteration (i.e., iteration $k-1$). The feature of the node *v* at the previous iteration ($f_v^{k-1}$) is added to the aggregation of adjacent nodes' features (i.e., the CONCAT function is a simple adding operation). Equation (2) shows the aforementioned feature extraction. $\Theta_k$ and $\Phi_k$ are the

trainable parameters of the proposed GNN. $\sigma$ indicates a nonlinear activation function, such as ReLU or sigmoid.

$$f_v^k = \sigma \left( \Theta_k \sum_{\{u|(u,v)\in E\}} \frac{f_u^{k-1}}{|\{u|(u,v)\in E\}|} + \Phi_k f_v^{k-1} \right) \tag{2}$$

The trainable parameters $\Theta_k$ and $\Phi_k$ are updated to minimize the loss function (3). The loss function of logistic regression is applied to evaluate the loss of the GNN. When two nodes $u$ and $v$ are connected, the cosine similarity approaches 1. Otherwise (i.e., $u$ and $v$ are disconnected), the cosine similarity is close to 0.

$$L = \sum_{(u,v)\in E} log\left(\sigma\left((f_u^k)^T\left(f_v^k\right)\right)\right) + \sum_{(u,v)\notin E} log\left(1 - \sigma\left((f_u^k)^T\left(f_v^k\right)\right)\right) \tag{3}$$

Actual node classification is performed using the trained GNN. The trained classification function combined with the GNN determines the node class and assigns tags (see spread/skip/activation in Figure 4). Note that a *softmax* classifier is typically used.

## 5. Experiments

### 5.1. Experiment Design

To demonstrate the superior performance of the proposed dynamic network configuration and dApp spreading, we built a test blockchain network in a cloud system. In our previous work [2], we reorganized and modified the Hyperledger sawtooth [40] blockchain software. The consensus engine, validator, database, and data-serialization functions were reorganized for small-scale IoT development devices. For convenience, we applied Docker [41] to the developed blockchain software module deployment. Docker is a container-based open-source virtualization platform. A Docker container is created from a Docker image, which contains the application to be run and its execution environment. Docker guarantees the same execution in various computer environments and can use services through images without complicated deployment steps [42]. We created a portable Docker image for easy distribution and maintenance of the developed blockchain software module. Figure 6a shows the Docker container installation to a virtual machine in the cloud system. We used the private cloud system supported by KOREN (KOrea REsearch Network). Our virtual blockchain testbed has a maximum of 100 virtual machines (VMs), each of which employs a Docker engine and a sawtooth-based blockchain software module (see Figure 6a). To emulate the network connectivity between VMs, we also deployed a pub/sub transmission between VMs. A transmitter VM publishes a transaction and a receiver VM subscribes to obtain the transaction. A broker node relays the pub/sub operations. The broker stores published transactions and sends transactions to subscribed receivers in each cluster. Note that the operational reliability of the broker is important to pub/sub operation. The instability of broker operation could cause message transfer failure. To prevent the broker failure, many popular pub/sub operation methods such as MQTT provide the data recovery method. After network clustering, a single VM is assigned to be a broker in each cluster. The position data of VMs and pub/sub connectivity demonstrates the blockchain network testbed in the cloud system. Figure 6b presents the visual diagram of the testbed. The NetAnim (network animation) simulator employs the network configurations generated from the testbed and shows the status of the VMs.

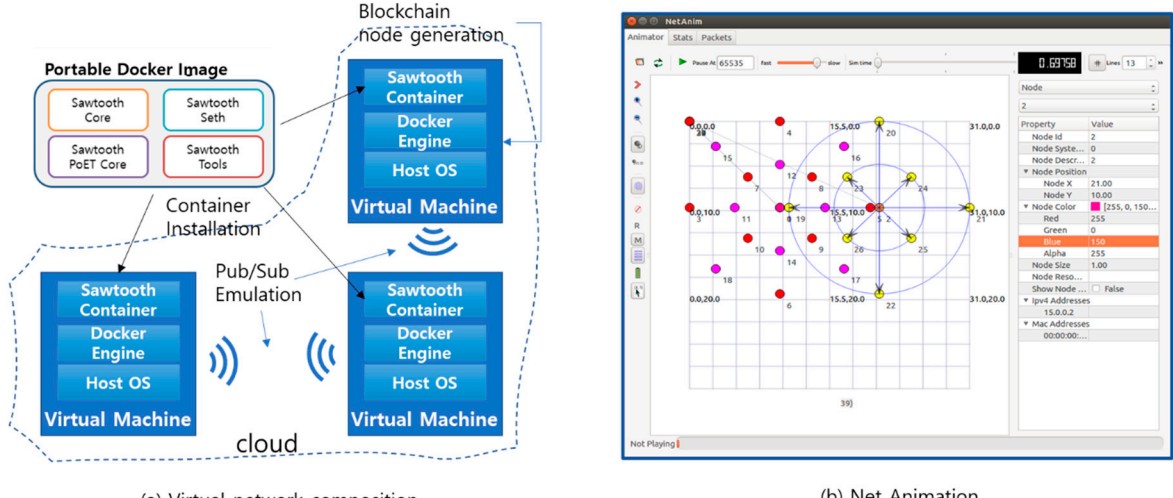

(a) Virtual network composition

(b) Net Animation

**Figure 6.** Blockchain node generation in cloud and visualization of network structure.

The VMs (i.e., network nodes in the blockchain network) are classified into two classes, comprising 90 sensor nodes and 10 actuator nodes. The transaction generation of nodes is also classified into two types: a reporting type with a normal distribution (mean 1/min, standard deviation 6/min) and a requesting type with another normal distribution (mean 6/h, standard deviation 10/h). The sensor nodes usually generate reporting transactions, and the actuator nodes generally generate requesting transactions. We assume that the destinations of the transactions are randomly selected. The network fluctuation is emulated by the node on/off. The most important network fluctuation is caused by membership changes in clusters. A randomly selected 10% of network nodes are turned off every hour (the turned-off node will be regenerated at the beginning of the next selection period) to emulate network fluctuation. In addition, we simulate node movement. Only a small number of network nodes move in real situations. Many sensing nodes, for example, are fixed to facilities or buildings. We randomly select 10% of VMs to emulate node movement. The moving nodes change their position at a speed of 1 m/h in random directions. The position data of moving nodes are continuously calculated and recoded to the node behavior tensor described in Figure 2a.

### 5.2. Results: Dynamic IoT Blockchain Configuration

First, we show the effect of the proposed deep clustering for IoT blockchain networks. We use a standard CNN architecture, which consists of five convolutional layers with 16, 32, 64, 32, and 16 filters, and two fully connected layers. Deep clustering is a sort of unsupervised learning methods. Thus, we do not need to label the network nodes for training. To train the CNN for deep clustering, we generate 24-h transactions with node on/off data as a single epoch. A total of 20 training iterations of epoch are performed to train the CNN. The test data are obtained from another 12-h set of transactions and node behaviors. Note that, the IoT node behavior model applied to the NS-3 network simulator. The NS-3 simulator can provide node behavioral data as the tensor form described in Figures 3 and 5. Figure 7 shows the effect of the proposed deep clustering method for lightweight blockchain networks. For comparison, we also illustrate successive *k*-means clustering with multiple iteration periods (1 h, 2 h, 3 h, 4 h). In the case of *k*-means with a 1-h iteration period, we apply *k*-means clustering every hour to the lightweight blockchain network.

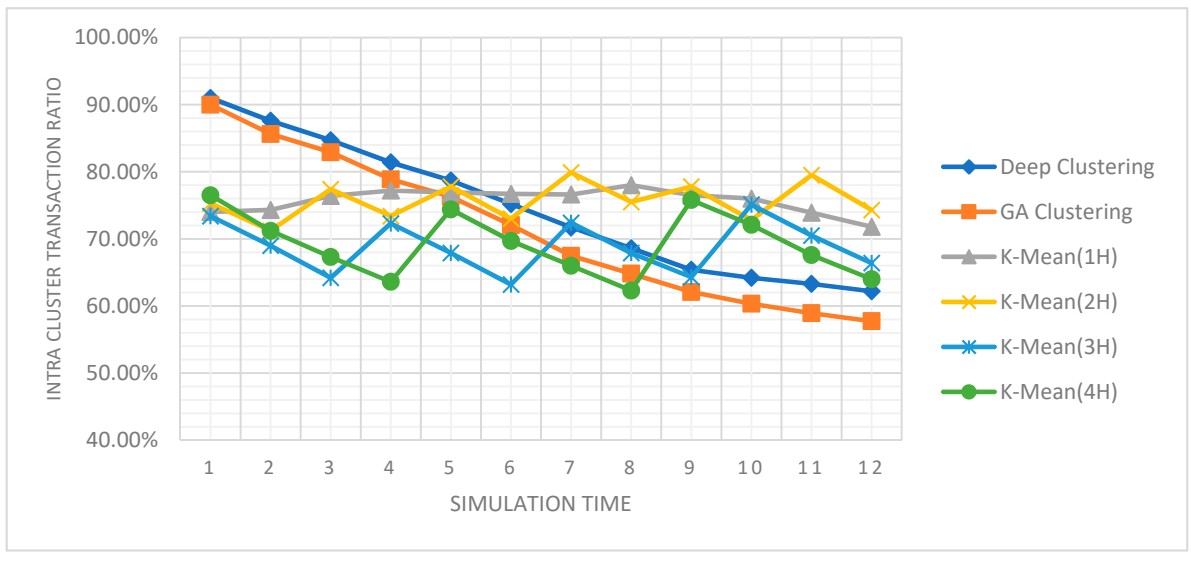

(**a**) Intra-cluster transaction ratio

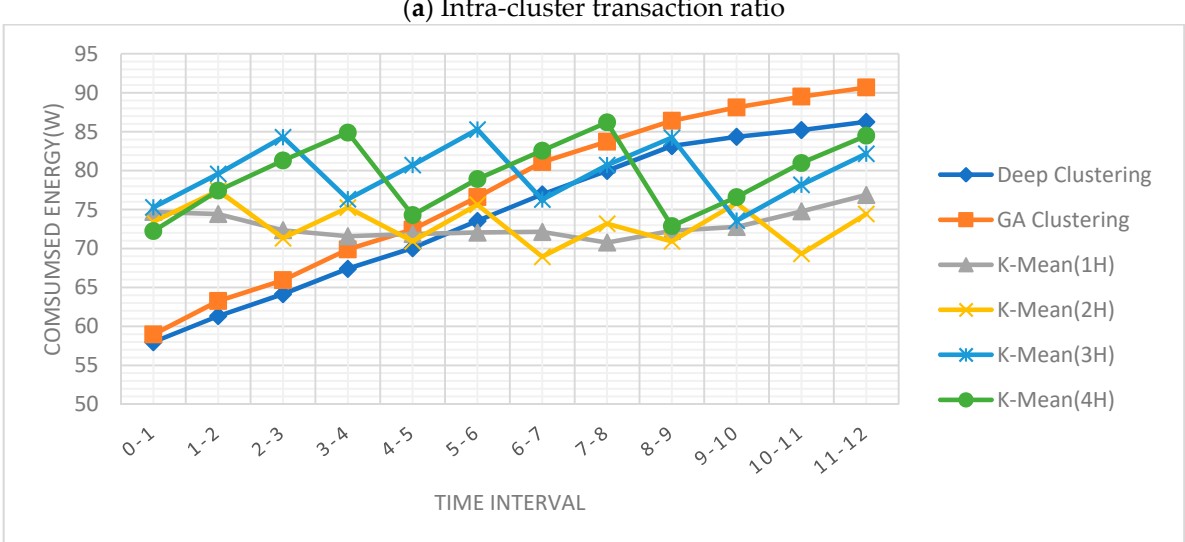

(**b**) Energy consumption in holistic network

**Figure 7.** Performance of proposed deep clustering in lightweight IoT blockchain network.

The first measure used to evaluate the proposed deep clustering is the intra-cluster transaction ratio. It is obvious that the transactions within a cluster require smaller transaction costs compared to those of inter-cluster transactions. The deep clustering method allows over 90% of transactions to be processed as intra-cluster transactions for the first hour of the test. The intra-cluster transaction ratio gradually decreases over time. At the end of the test, the intra-cluster transaction ratio is close to 60%. Applying only single deep clustering, we observe that over 60% of transactions are processed inside the cluster for 12 h. The successive *k*-means clustering has a relatively low intra-cluster transaction ratio at the beginning of the test (i.e., roughly 75% or less). The successive application of *k*-means clustering prevents excessive performance degradation. For example, applying *k*-means clustering every 3 h (i.e., *k*-means (3 h) in Figure 7a) restores the highest intra-cluster transaction ratio for every 3 h. However, repeated clustering requires extra data gathering and calculation of a central unit. Practically, repeated clustering should be avoided to maintain network configuration stability. Figure 7b estimates the energy consumption of transactions in the network. We assume WiFi connectivity between nodes. A report from Gomez et al. [43] illustrated the energy consumption per WiFi message transmission. We assume that the transmission power of the network node is 10 dBm with 10 mW needed

to publish a single transaction (we fix the transmission speed to 100 kbps, which is sufficient for sensing or actuating IoT nodes). In our experiment, all IoT nodes are located within a circle with 100 m radius. To simulate both of intra-cluster and inter-cluster transactions, 100 m radius is enough to consist multiple clusters. An intra-cluster transaction requires single-transaction publishing to the broker in the cluster. The transmission energy of a single message is sufficient for intra-cluster transactions. However, inter-cluster transactions require multiple transmissions. The published transaction should be relayed over the brokers. The broker of the published side relays the transaction to the broker of the subscribed side.

Figure 7 also illustrates the performance of dynamic clustering based on the genetic algorithm (GA) [44]. To optimize the clustering with historical data of network and IoT node behaviors, they built initial clustering solutions and try to converge to the optimal clustering. The input node behaviors and suggested fitness function are similar to the two-dimensional input tensor form and the loss function of our proposed method. The energy consumption and the ratio of intra-cluster transactions illustrated in Figure 7 are similar to the proposed deep clustering. However, the converging time of GA is greater than the deep clustering. GA needs a relatively long time to converge to the final solution (the minimum converging time is 2600 ms and the maximum is measured at 4300 ms). Because of the slow convergence of GA to find the final solution, the GA clustering is hard to apply the practical clustering in a real-time fashion. Note that, the convergence time is measured in 1.2 Ghz CPU and 1GB RAM of VM. The specified CPU and RAM capability is the popular specifications of RasberryPi3.

### 5.3. Discussion: Dynamic IoT Blockchain Configuration

Using deep clustering, we can build the cluster configuration for the IoT blockchain network and then classify the intra- and inter-cluster transactions. The energy consumption patterns illustrated in Figure 7b show the inverse form of the patterns in Figure 7a (i.e., the patterns of intra-cluster transaction ratios). This reciprocal relationship is natural because a high intra-cluster transaction ratio guarantees large energy savings for transaction publishing.

### 5.4. Results: Reduced dApp Spreading using GNN Node Classification

Here, we show the effect of the proposed GNN for IoT blockchain networks. To train the GNN for node classification (i.e., assigning node tags such as spread/skip/activation), we prepared two types of dApps: concurrent sensing transaction publishing and serial sensing transaction publishing (see Figure 8).

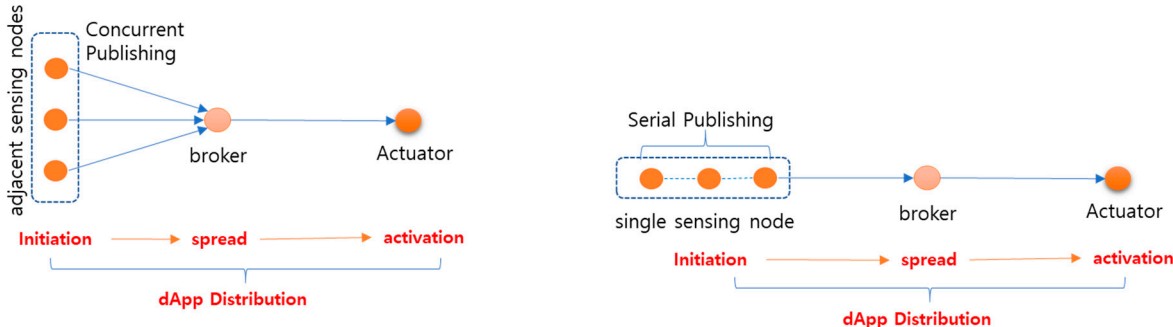

(**a**) Concurrent sensing transaction publishing    (**b**) Serial sensing transaction publishing

**Figure 8.** Smart contract types of node classification.

The concurrent sensing-type dApp is initiated by a set of adjacent sensor nodes. When the adjacent sensor nodes detect the same situation, they generate concurrent sensing transactions, and then an actuator node starts its action. The broker processes and relays the published concurrent sensing transactions to the actuator node. For example, high temperature measured by adjacent sensor nodes in a factory activates an alarm and air conditioning systems. The serial sensing-type dApp is initiated by a single sensor node. When a single sensor node detects the same situation over a predetermined time interval, actuation is performed in the actuator. The broker has the same role as in concurrent sensing-type transactions. An example of a serial sensing-type dApp is the detection of machine malfunctions. A single error detection is not sufficient to determine the malfunction; multiple consecutive error detections are required. For GNN parameter training, fifty concurrent sensing types and the same number of serial sensing-type dApps are used in our experiment. The adjacent node sets and their actuators are randomly selected for each concurrent sensing-type dApp. A single sensing node and its actuator are also randomly selected for each serial sensing-type dApp. The information of node status is gathered from the monitoring module of each IoT node. The sawtooth-based blockchain software employed in the VMs has a monitoring module to report status of IoT nodes. After the selection of sensor and actuator nodes, the broker nodes can be determined according to the cluster configurations. The broker belonging to the same cluster as the sensor nodes relays the sensing transactions and employs the dApp.

To prove the effectiveness of the proposed GNN method for node classification, we generate 1000 transactions that initialize the dApps to actuators. Of these transactions, 50% initiate concurrent sensing-type dApps and the other 50% initiate serial sensing-type dApps. The transactions cause changes in the behavior tensor of the nodes $x_v$; then, the trained GNN determines the dApp spreading to the actuators and brokers. The performance of the proposed GNN can be measured by the failure of dApp spreading. Incorrect spreading of the dApp necessitates impending dApp downloads to brokers and actuators. These impending downloads should be avoided. The delay caused by these downloads can easily lead to the failure of dApp execution. The flooding of dApps across the network is the basic operational process to avoid execution failure. However, broad proliferation requires excessive memory for IoT nodes, imposing a heavy burden on memory-constrained IoT nodes. Our GNN-based node classification for dApp spreading guarantees the successful execution of dApps while minimizing the unnecessary excessive proliferation of dApps. Figure 9a shows the frequency of impending downloads for brokers and actuators, which are caused by incorrect node classification and dApp spreading. A total of 1000 transactions are tested. Each transaction initiates the dApp. The dApps should be activated through the spreading line (both the relaying brokers and actuator). The failures of node classification (i.e., causing impending dApp downloads for a broker or actuator) are restricted to 3%–5% for every transaction interval (we divide 1000 transactions into 10 intervals, where each interval has 100 transactions). Note that the impending downloads for both the broker and actuator have similar patterns. The impending downloads for both are restricted to very small portions.

Figure 9b shows the memory requirements of each IoT node. Because of the characteristic feature of the blockchain, dApp flooding involves a large memory requirement for every network node.

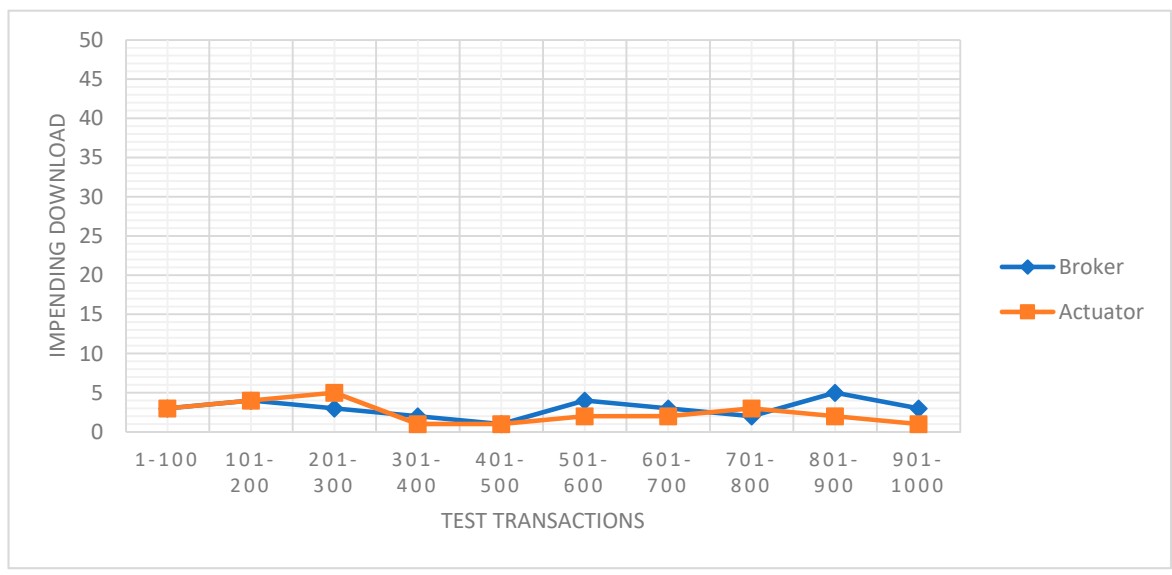

(**a**) Impending downloads for broker and actuator

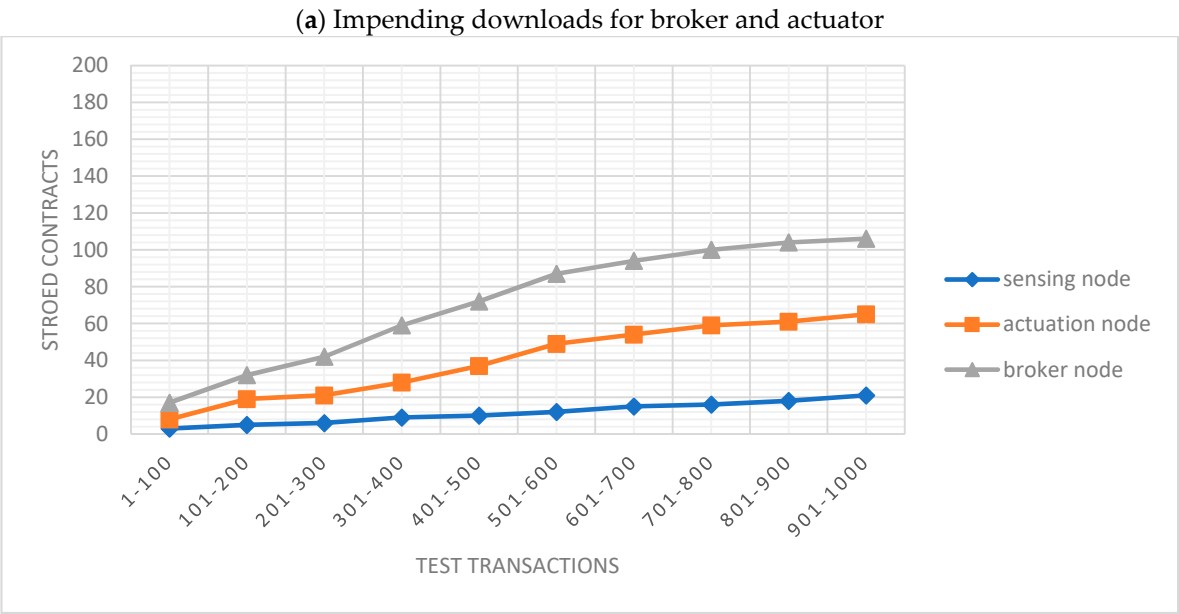

(**b**) Stored smart contracts

**Figure 9.** Performance of proposed GNN node classification in lightweight IoT blockchain network.

### 5.5. Discussion: Reduced dApp Spreading Using GNN Node Classification

Theoretically, every network node must store all dApps (i.e., 1000 dApps for this test) to ensure successful operation between sensors and actuators. The GNN-based node classification for dApp spreading dramatically reduces the memory requirement. Sensor nodes store only 21 dApps, actuation nodes have under 65 dApps, and broker nodes store only 106 dApps to handle the total 1000 test transactions. In addition, Figure 9b shows that the volume of stored dApps increases gradually until the middle section (i.e., 601–700 transaction interval) and then converges to a plateau. This observation suggests that the increase in stored dApps can be restricted to a certain level, even for continuous transaction generation.

### 5.6. Data Security Validataion

To test the reliability of the proposed methods, we add the attack agent that generates the fake transactions. The attack agent monitors the status of IoT blockchain node and

determines the fake transaction generation. The status of the node is described by the node reputation and block validity. The node reputation is measured by the number of blocks the node has generated so far. Usually, the more block generation means the higher reputation of the node. Block validity is determined by the number of block-approved nodes. When a block is generated and spreads to the network, the member nodes of the network check the validity of the block and approve the block. The more nodes that have approved the block, the higher the validity of the block. The attack agent has a function for fake transaction generation.

$$f = rand\left(0, ae^{-b(r \times h)}\right) \tag{4}$$

The $f$ denotes the number of generated fake transactions. $r$ means the node reputation and $h$ means block validity. $a$ and $b$ denote the constants that control the maximum level of fake transaction generations and the composite effect of $(r \times h)$. $f$ is randomly selected between 0 and $ae^{-b(r \times h)}$. We apply the attack agent to the selected IoT nodes in the tested network. Because of the robust consensus algorithm (i.e., Proof of Elapsed Time) and the Merkle tree validation of the sawtooth blockchain, the fake transactions cannot be included in the generated blocks. One hundred percent of fake transactions are detected and excluded in the transaction validation process of sawtooth consensus. However, if the attack agents penetrate above the critical points, for example over 1/3 of all nodes, the network can be vulnerable from fake transactions. (Practical Byzantium Fault Tolerance consensus can guarantee the fault-free if we have 2/3 correctly working nodes. An agreement on block validation can be reached with 2/3 validation of correctly working nodes). The data security and reliability essentially depend on the blockchain itself. The cross validation of blocks, Merkle tree, consensus, and encryption keys of blockchain provide the network security. The clustering or node classification proposed from our research work aims to the efficient operation of the IoT blockchain network while keeping the flawless reliability of blockchain. The embedded sawtooth software module faithfully guarantees the transaction security and zero tolerance of blockchain.

## 6. Threats to Validity

The essential limitation is the limited validation of proposed works. We evaluate the performance of proposed methods under the general network environment. We defined experimental setup including IoT node classes, transaction types, transaction generation patterns, network fluctuation caused by node movement, and dApp classes. This experimental setup illustrates a normal IoT network environment. However, we must apply to various situations to prove the completeness of the proposed methods. Because of the computing power limitation of the tested simulation system and insufficient flexibility of VM operations, we cannot have the experimental diversity. Another limitation is caused by VM. The VMs are operated over the private cloud system supported by KOREN. Because of the limited budget to employ the blockchain software module to real IoT nodes, we built homogeneous VMs over the cloud system. The homogeneous VM cannot reflect the actual operations of various IoT nodes. Moreover, the number of VMs is limited to 100 nodes for the experiments. We could not obtain sufficient VMs from the cloud system provider.

## 7. Conclusions

Machines need to be connected, including vehicles, robots, drones, home appliances, displays, smart sensors installed in various infrastructures, construction machinery, and factory equipment. A tremendous amount of data associated with hundreds of billions of connected machines and humans needs to be collected and utilized for advanced user services. To accomplish this, artificial intelligence will need to be embedded in all system components. The native AI allows all system components to obtain and evaluate an enormous amount of online/offline data. The massively connected devices and data will increase the openness of communication networks and hence increase the attack surface. This could make the entire system more vulnerable to security and privacy threats. The lightweight blockchain software module can be applied to common IoT devices;

its practical applicability in this context is guaranteed by its enhanced modular architecture and lightweight consensus mechanism. Direct embedding of the lightweight blockchain middleware module in small computing devices was proved to be practicable in our former works. The highly compatible blockchain software platform ensures end-to-end secure transaction transfer and eliminates the oracle risk of a typical blockchain platform.

However, the development of a highly compatible blockchain software module is not sufficient for practical networks. Effective configuration of blockchain networks and operation procedures are particularly important for actual service provision. We have proposed dynamic network clustering and node classification for blockchain network deployment and operation. The proposed deep clustering builds iterative clusters for IoT blockchain networks. A two-dimensional tensor is suitable to record the time-varying behaviors of each network node, and the feature extractor uses a network behavior snapshot as input data. A feature vector obtained from the extractor is used to build network node clusters. GNN-based node classification guarantees optimized dApp spreading. This approach can significantly improve the processing speed during the transaction verification process. We propose a spreading method for dApps that can be linked with dynamic clustering in an IoT environment. In the network cluster, a software agent powered by artificial intelligence assigns tags to nodes according to their states and computing loads. Deep clustering and GNN are originally proposed to apply for object classification. The original deep clustering was invented for image classification. The CNN structure embedded in deep clustering is best suitable for image data that has a two-dimensional tensor form. The compactness and fast computing of CNN structure are fully beneficial to the higher performance for image classification of the deep clustering. The proposed time-varying two-dimensional tensor that records whole network behavior expands the applicability of deep clustering to IoT blockchain node clustering. The expanded coverage of deep clustering and the two-dimensional tensor structure for network behavior representation imply the theoretical advance of proposed works. The usability of deep clustering can be drastically penetrated to various application fields. The computation results of IoT node clustering illustrate the practical applicability of deep clustering to IoT blockchain networks. The proposed deep clustering derives better robustness compared to successive k-means clustering and GA (Genetic Algorithm)-based clustering. Even a single applying of deep clustering surpasses the repetitive execution of traditional k-means clustering.

The iterative feature updates by GNN derive the advantage for node classification. The GNN has aggregation and concatenation functions to extract the feature vectors of nodes. The input behavior tensor of a node is concatenated with its adjacent nodes' aggregated behaviors. To enhance of whole blockchain network performance, each network node should have a specific role for the dApp distribution. The original GNN has the advantage to identify the object relationship in static social graphs. The proposed node classification by GNN also expands the usability of GNN to dynamic communication networks. The practical implication of the proposed GNN-based IoT blockchain node classification is illustrated from the computational results. The very low impending downloads of dApp and the reduced memory requirement for each IoT node prove the practical superiority of GNN-based node classification.

The proposed clustering and node classification using the lightweight blockchain module can be easily applied to various blockchain networks. Our lightweight blockchain deployment and operation is most applicable to massive IoT networks, one of the essential 5G network structures. Various small IoT devices embedded in a lightweight blockchain module can provide very stable and efficient services using the proposed blockchain network configuration and operation procedures.

**Author Contributions:** Conceptualization, J.-H.K. and S.H.; methodology, J.-H.K.; experiment, S.L.; validation, S.H., writing—original draft preparation J.-H.K.; writing—review and editing, S.H. All authors have read and agreed to the published version of the manuscript.

**Funding:** This work was supported in part by a grant from the Institute for Information and Communications Technology Promotion (IITP) funded by the Korean Government (Ministry of Science and Information Technology) (Versatile Network System Architecture for Multi-Dimensional Diversity) under Grant 2016000160, and in part by the National Research Foundation of Korea (NRF) grant funded by the Korean Government (Ministry of Science and Information Technology) under Grant 2020R1F1A1049553.

**Conflicts of Interest:** The authors declare no conflict of interest.

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
