# Peer review of "Autonomous Operation Control of IoT Blockchain Networks"

_electronics, doi:10.3390/electronics10020204_

Round 1

Reviewer 1 Report

The authors proposed an essential operational advances to develop a lightweight blockchain over IoT networks using artificial intelligence.Although the paper is well written with technical sound, I have the following comments:

-- why the choice of neural network technique than other machine learning algorithm? please justify

-artificial neural network  is more powerful, why did you opt for CNN instead

-- How data were gathered? 

--- How the features were extracted?

-- What are the limitations of your work? 

-- Some references need to be updated in order to reflect the novelty of the proposed work (1967 2003, 2005, .... )

--- The authors need to clearly identify theoretical and practical implications of their research

Author Response

Thank you for kind suggestion, 

We prepare the answers and revised version carefully.

Reviewer 2 Report

Paper presents a relevant topic and indeed, enabling blockchain adoption in the current IoT system can potentially increase their reliability and security, topics, that are frequently raised as concerns in the IoT field.

I would suggest improving the manuscript in several points I list in my review and encourage authors to submit a revised version of the paper.

Abstract:  “Internet of Things (IoT) networks are composed of many sensors and actuators” I would suggest “typically composed of…”

Line 52:  I think mentioning reliability and security concerns in current IoT is an important point to be done here. I would suggest strengthening the discussion and support it by more recent and extensive studies in the field. Some examples that might be useful:

Ahmed, B. S., Bures, M., Frajtak, K., & Cerny, T. (2019). Aspects of quality in Internet of Things (IoT) solutions: A systematic mapping study. IEEE Access, 7, 13758-13780.

Khan, M. A., & Salah, K. (2018). IoT security: Review, blockchain solutions, and open challenges. Future Generation Computer Systems, 82, 395-411.

I would suggest adding a separate section dedicated to related work; now this part is distributed in the introduction and following chapters; however, it would be appropriate to have related work section in a journal paper.

Figures 7 and 9 – adding vertical gridlines would be helpful in reading of the data from the graph.

Generally, the spacing between figures and text as well as the text and given formulas have to e improved; an example are lines 191 – 199 – the formatting makes the text harder to read.

Split the section Experiments to more subsections allowing better readability—a suggestion: experiment design, results, discussion.

Add a “threats to validity” section discussion possible bias in the experimental method and acquired data, as well as possible limits and drawbacks of the proposed concept.

I would suggest extending the conclusion by more extensive summary using the main figures from the experiments you have conducted.

Let the paper proofread by a native speaker to achieve a better level of English.

I hope my comments help and working them in will contribute to improving the manuscript.

Author Response

(The authors gave the same response as above.)

Reviewer 3 Report

The manuscript is interesting and may provide good insights to solve the issues related to the lack of hardware resources in standard IoT networks. Indeed, this usually prevents IoT network from implementing an effective block-chain architecture to guarantee security (authenticity) and reliability of transactions. However, I have some concerns, mainly related to the assessment process, that need to be addressed.

1. Is 100 a reasonable number for block chain nodes? It seems far too low.

2. Is reliability (and liability) of broker node relaying pub/sub operations a matter of concern?

3. It looks like correct behaviour of network considered here as an example (90 sensors, 10 actuators) may depend on physical locations of nodes, thus their clustering. Are there any constraints related to node physical location (for instance, their physical distance?).

4. While the assessment of the requirements for the considered IoT network may lead to promising results, the lack of any comparison does not allow the reader to fully appreciate the validity of the proposed clustering approach.

5. Additionally, since the main objective of a blockchain is to provide data security and reliability, I believe that this aspect has not been fully evaluated. Proper metric for this evaluation should be proposed and assessed, and compared with those of alternative approaches.

Author Response

(The authors gave the same response as above.)

Round 2

Reviewer 2 Report

Authors have satisfactorily reflected my comments, hence I suggest accepting the manuscript to be published.

Some small issues to fix:

Line 59: “The IoT network has considerable operation instability” … I don’t think this is generally the case for all IoT networks; some of them can be relatively stabile

Line 103:  a typo:  illustrate the reliability…

Fig 2 and Fig 3:  Blue squares can be drawn by different color to allow better readability

485:  Treats to validity is usually organized as a separate section before the Conclusion

Author Response

Thank for kind comments.

We write the answers for comments. 

Reviewer 3 Report

I would like to thank the Authors for addressing my concerns. I am happy with the modifications introduced in the revised version.

I would just recommend the Authors to carefully readproof the manuscript before submitting their final version. There are a few paragraphs that do not read very well.

Author Response

(The authors gave the same response as above.)
